# Depressive symptoms and associated factors among HIV positive patients attending public health facilities of Dessie town: A cross-sectional study

**Yitayish Damtie**[1]*, **Bereket Kefale**[1], **Melaku Yalew**[1], **Mastewal Arefaynie**[1],
**Bezawit Adane**[2], **Afework Edmealem**[3], **Atsedemariam Andualem**[3]

1 Department of Reproductive and Family Health, School of Public Health, College of Medicine and Health Sciences, Wollo University, Dessi, Ethiopia, 2 Department of Epidemiology and Biostatistics, School of Public Health, College of Medicine and Health Sciences, Wollo University, Dessi, Ethiopia, 3 Department of Adult Health Nursing, School of Nursing and Midwifery, College of Medicine and Health Science, Wollo University, Dessi, Ethiopia

* yitutile@gmail.com

**Data Availability Statement:** All relevant data are within the manuscript and its Supporting information files.

## Abstract

### Background

Depressive symptoms are the most common psychiatric complication of Human Immunodeficiency Virus (HIV) infection. They are associated with poor drug adherence, treatment failure, and increase the risk for suicide. There was limited evidence of depressive symptoms among HIV-positive patients in the study area. So, this study aimed to determine the prevalence of depressive symptoms and associated factors among HIV-positive patients attending public health facilities of Dessie town, North-central Ethiopia, 2019.

### Method

A cross-sectional study was conducted on 380 HIV-positive patients attending ART clinics in Dessie town, North-central Ethiopia, 2019. Samples were selected using systematic random sampling and the data were collected by using structured, pretested, and interviewer-administered questionnaires. Patient Health Questionnaire (PHQ-9) at a cut-off point of 5 was used to assess depressive symptoms. The data were entered by Epi data version 3.1 and analyzed by SPSS version 25. A binary logistic regression model was used to identify factors associated with depressive symptoms. The Adjusted Odds Ratio (AOR) along with a 95% Confidence Interval (CI) was estimated to measure the association. The level of significance was declared at a *p*-value of less than 0.05.

### Result

The prevalence of depressive symptoms among HIV positive patients was 15.5% (95% CI: (12.4%, 19.2%)). Age 40–49 years compared to 30–39 years (AOR = 2.96, 95% CI: (1.01, 8.68)), age ≥50 years compared to 30–39 years (AOR = 3.81, 95% CI: (1.05, 13.8)), having perceived stigma (AOR = 10.2, 95%CI: (4.26, 24.4)) taking medication other than

**Funding:** The author(s) received no specific funding for this work.

**Competing interests:** The authors have declared that no competing interests exist.

**Abbreviations:** AOR, Adjusted Odds Ratio; ART, Antiretroviral Therapy; CD$_4$, Cluster Differentiation$_4$; CI, Confidence Interval; ERC, Ethical Review Committee; HADS, Hospital Anxiety Depression Scale; HAM-D, Hamilton Depression Rating Scale; HIV, Human Immunodeficiency Virus; HIV/AIDS, Human Immunodeficiency Virus/ Acquired Immune Deficiency Syndrome; MADRS, Montgomery Asberg Depression Rating Scale; PHQ-9, Patient Health Questionnaire-9; SDG, Sustainable Development Goal; SPSS, Statistical Package for Social Science; SSA, Sub Saharan Africa; UNAIDS, Joint United Nations Programme on HIV/AIDS; WHO, World Health Organization.

Antiretroviral Therapy (ART) (AOR = 2.58, 95% CI: (1.25, 5.33)) and history of opportunistic infections (AOR = 5.17, 95% CI: (1.31, 20.4)) were factors associated with depressive symptoms.

## Conclusion

The prevalence of depressive symptoms was low compared to previous studies. Age, perceived stigma, taking medication other than ART, and history of opportunistic infections were factors associated with depressive symptoms. Health education and counseling programs should be strengthened and target older patients, patients who took medications other than ART, patients who experienced perceived stigma and patients with a history of history opportunistic infections.

## Introduction

HIV has become a major health challenge for the global population. Currently, it affects more than 37.9 million world population and 70% of patients were from sub-Saharan African (SSA) countries [1, 2]. In Ethiopia, it was estimated in 2018 that 399,000 of the urban population were living with HIV/AIDS [3].

Depressive symptoms are the fourth leading cause of disability affecting 350 million people worldwide [4, 5]. It disproportionately affects people living with HIV, with a recent meta-analysis indicating a two-fold higher risk compared to the general population [6]. Sub-Saharan African countries are highly affected by depressive symptoms in which 3.63 million people living with HIV (PLHIV) have major depressive symptoms [7]. In Ethiopia, previous studies assessing the prevalence of depression among PLWH have differed considerably, with estimates ranging from 11.7–76.7% [8–12].

Depressive symptoms impose a significant adverse effect on HIV-positive patients. They affect the patient's ability to adhere to their ART medication. A recent study showed that HIV patients with depressive symptoms were 3.3 times more likely to be non-adherent to their ART medication as compared to their counterparts [13]. They also lead to ART treatment failure, drug resistance, and early disease progression as a result of poor adherence to ART medication [14, 15]. Depressive symptoms affect a patient's quality of life, it decreases life expectancy and increases suicidal ideation. They have an association with risky sexual behavior which increases the risk of HIV transmission [16–21].

Depressive symptoms have been associated with a wide range of socio-demographic factors, including age, sex [11], lower socio-economic class [11, 22], unemployment [22] and living alone [8]. In addition, clinical factors like HIV status non-disclosure [8], advanced World Health Organization (WHO) clinical stage [9, 11], presence of stigma [8, 10, 23–25], adverse drug reaction [9, 24], presence of opportunistic infection [24], low Cluster Differentiation4 (CD4) count [11] and presence of comorbid illness [24] were another groups of factors affecting depressive symptoms.

The Sustainable Development Goals (SDGs) particularly Goal 3 aims to end the HIV/AIDS epidemic by 2030 [26]. The Joint United Nations Program on HIV/AIDS (UNAIDS) also plans to achieve 90-90-90 targets by 2020 [27]. So, Intervening factors linked to depression is an essential strategy to achieve these two major international commitments as it leads to poor ART drug adherence and further transmission of HIV/AIDS.

There was limited evidence on depressive symptoms among HIV-positive patients in Dessie town. The finding of this study could be an input to provide evidence-based intervention to prevent depressive symptoms among HIV-positive patients. Thus, this study aimed to determine the prevalence of depressive symptoms and associated factors among HIV-positive patients in Dessie town.

## Materials and methods

### Study area, study design, and participants

A cross-sectional study was conducted from March 1-30/2019 in the public health facilities of Dessie Town, which is a city located within North-central Ethiopia. Dessie is the center of the south Wollo Zone located 401KM away from Addis Ababa, the capital city of Ethiopia and 480 km away from Bahir Dar. According to the 2019 Dessie town administrative health office report, two public hospitals, and three health centers were providing ART services for a total of 9953 adult HIV-positive patients. The study population was randomly selected from all adults with HIV attending public health facilities of Dessie town. Adult HIV-positive patients greater than or equal to 18 years old and who were on ART for at least six months were included in this study.

### Sample size and sampling procedure

The study sample size was calculated for the primary risk factor as presence of antiretroviral drug side effects, considered as a binary variable. Based on previous research [9], we wished to detect differences in adjusted odds ratios of at least 4.7, assuming an underlying prevalence of 10.1%, power of 80% and 5% Type I error. After adjustment for 10% non-response, the minimum required sample size was 388. Systematic random sampling was used to select participants using the antiretroviral drug dispensary registration book as the sampling frame. Study participants were invited to participate when attending the health facility for antiretroviral drug collection.

### Data collection procedures and measurements

Data were collected by face-to-face interviews using a structured and pretested Amharic version questionnaire. The patient's medical record was also reviewed to identify clinical markers like CD4 count and WHO clinical staging. Three trained nurses have collected the data with the supportive supervision of the principal investigator and one public health officer supervisor. The questionnaire collected data on socio-demographic, clinical, psycho-social, and behavioral characteristics [9, 11, 22–25, 28, 29].

Depressive symptoms were assessed using the PHQ-9 scale. The PHQ score ranged from 0 to 27, a score of 0–4, represents no depression and a score of 5–9, 10–14, 15–19 and 20–27 represent mild, moderate, moderately severe and severe depressive symptoms respectively. In this study, PHQ score $\geq 5$ was used as a benchmark to categorize depression. Accordingly, patients were said to have depressive symptoms if they scored $\geq 5$ and have no depressive symptoms if they scored <5 [24, 29].

Perceived stigma was measured using ten questions with 'yes' or 'no' response options. These questions were adapted from the HIV stigma index validation survey conducted in six Iranian cities. It has been validated in our country as well (in Iimma town) and hasn't a cutoff point like the PHQ-9 depression scale. The overall score of perceived stigma was calculated out of ten and the distribution was checked for normality. Since the data was not normally distributed, we use the median value to classify perceived stigmas as 'yes' or 'no'. So, patients who

scored above the median score (>0) were considered as they had perceived stigma whereas patients who scored less than or equal to the median score (≤0) had no perceived stigma [30].

## Statistical analysis

Data were coded and entered into Epi Data Version 3.1 and analyzed by using Statistical Package for Social Science (SPSS) Version 23 statistical software. Descriptive statistics like mean, median, frequency, and proportion were computed. The association between independent variables and depressive symptoms was made using a binary logistic regression model and a $p$-value less than 0.2 was used to screen eligible variables for multivariable analysis. Hosmer and Lemeshow goodness of test and standard error were used to check the model fitness and Multicollinearity respectively. Variables having an adjusted odds ratio with a 95% confidence interval not inclusive of the null value were considered as associated factors of depressive symptoms. Statistical significance was declared at a $p$-value less than 0.05.

## Ethical approval

Ethical clearance was obtained from the Ethical Review Committee (ERC) of Wollo University. A permission letter to conduct the study was obtained from the school of public health and it was submitted to ART clinics of Dessie town. After explaining the objective of the study, verbal informed consent was obtained from each study participant. They were informed about their rights to withdraw from the interview at any time. Patients who were assumed to have depressive symptoms were linked to the psychiatric department for investigation and treatment. The information they provide was used only for the study purpose.

This manuscript was organized and written according to strengthening the Reporting of Observational Studies in Epidemiology (STROBE) 2007 (v4) Statement checklist for cross-sectional studies (S1 Table).

## Results

### Socio-demographic characteristics

In this study, a total of 380 patients were participated making a response rate of 97.9%. The median age of the study participants was 36 years with an interquartile range of 12 years. The overall prevalence of depressive symptoms among HIV-positive patients was 59(15.5%). Twenty-eight (47.5%) HIV-positive patients aged 18–29 years and 7 (11.9%) patients aged greater than or equal to 50 years old had depressive symptoms. The prevalence of depressive symptoms among female patients was 66.1%. Depressive symptoms were higher among married patients (54.2%) and patients who live in urban areas (78%) compared to their counterparts. Forty-six (78%) study participants who live with their families had depressive symptoms compared to 13 (22%) those who live alone (Table 1).

### Clinical, psycho-social and behavioral characteristics

Forty-one (69.5%) patients who took medication other than ART had depressive symptoms as compared to 30.5% of patients who didn't take medication other than ART. Similarly, 44 (74.6%) patients who disclosed their HIV serostatus to anyone else had depressive symptoms compared to 15 (25.4%) patients who didn't disclose their HIV serostatus to somebody else. Of a total of 114 (30.0%) study participants who experienced perceived stigma, 42 (71.2%) of them had depressive symptoms (Table 2).

**Table 1. Socio-demographic characteristics of HIV positive patients attending public health facilities of Dessie town, Northeast Ethiopia, 2019.**

| Variable | Depression symptoms (n = 380) | | | $X^2$, df |
|---|---|---|---|---|
| | Yes | No | Total | |
| Age | | | | |
| 18–29 years | 28 (47.5%) | 85 (26.5%) | 113 (29.7%) | 13.730, 3 |
| 30–39 years | 14 (23.7%) | 142 (44.2%) | 156 (41.1%) | |
| 40–49 years | 10 (16.9%) | 68 (21.2%) | 78 (20.5%) | |
| $\geq$50 years | 7 (11.9%) | 26 (8.1%) | 33 (8.7%) | |
| Sex | | | | |
| Male | 20 (33.9%) | 143 (44.5%) | 163 (42.9%) | 2.308, 1 |
| Female | 39 (66.1%) | 178 (55.5%) | 217 (57.1%) | |
| Marital status | | | | |
| Single | 27 (45.8%) | 140 (43.6%) | 167 (43.9%) | 0.093,1 |
| Married | 32 (54.2%) | 181 (56.4%) | 213 (56.1%) | |
| Residence | | | | |
| Urban | 46 (78%) | 254 (79.1%) | 300 (78.9%) | 0.040, 1 |
| Rural | 13 (22%) | 67 (20.9%) | 80 (21.1%) | |
| Ethnicity | | | | |
| Amhara | 57 (96.6%) | 289 (90%) | 346 (91.1%) | 2.648, 1 |
| Others[a] | 2 (3.4%) | 32 (10%) | 34 (8.9%) | |
| Religion | | | | |
| Christian | 26 (44.1%) | 135 (42.1%) | 161 (42.4%) | 0.083, 1 |
| Muslim | 33 (55.9%) | 186 (57.9%) | 219 (57.6%) | |
| Level of education | | | | |
| Have no formal education | 16 (27.1%) | 74 (23.1%) | 90 (23.7%) | 7.518, 3 |
| Grade1-8 | 18 (30.5%) | 131 (40.8%) | 149 (39.2%) | |
| Grade9-12 | 12 (20.3%) | 82 (25.5%) | 94 (24.7%) | |
| College and above | 13 (22%) | 34 (10.6%) | 47 (12.4%) | |
| Occupation | | | | |
| Government employed | 13 (22%) | 38 (11.8%) | 51 (13.4%) | 8.412, 4 |
| Private employed | 10 (16.9%) | 45 (14%) | 55 (14.5%) | |
| Housewife | 10 (16.9%) | 42 (13.1%) | 52 (13.7%) | |
| Farmer | 11 (18.6%) | 59 (18.4%) | 70 (18.4%) | |
| Others[b] | 15 (25.4%) | 137 (42.7%) | 152 (40%) | |
| Living condition | | | | |
| Family | 46 (78%) | 284 (88.5%) | 330 (86.8%) | 4.816, 1 |
| Alone | 13 (22%) | 37 (11.5%) | 50 (13.2%) | |
| Wealth index | | | | |
| Poorest quintile | 19 (32.2%) | 62 (19.3%) | 81 (21.3%) | 9.849, 4 |
| Poorer quintile | 14 (23.7%) | 81 (25.2%) | 95 (25.0%) | |
| Middle quintile | 5 (8.5%) | 36 (11.2%) | 41 (10.8%) | |
| Richer quintile | 8 (13.6%) | 92 (28.7%) | 100 (26.3%) | |
| Richest quintile | 13 (22%) | 50 (15.6%) | 63 (16.6%) | |

[a] Tigray and Oromo;

[b] student, merchant, daily laborer and jobless.

## Prevalence of depression

In our study, the prevalence of depressive symptoms among HIV-positive patients was 59 (15.5%) with a 95% CI: (12.4%, 19.2%).

**Table 2. Clinical, psycho-social, and behavioral characteristics of HIV positive patients attending public health facilities of Dessie town, Northeast Ethiopia, 2019.**

| Variable | Depression (n = 380) | | | $X^2$, df |
|---|---|---|---|---|
| | Yes | No | Total | |
| Number of ART pills taken per day | | | | |
| 1 | 27 (45.8%) | 124 (38.6%) | 151 (39.7%) | 1.169, 2 |
| 2 | 11 (18.6%) | 74 (23.1%) | 85 (22.4%) | |
| 3 | 21 (35.6%) | 123 (38.3%) | 144 (37.9%) | |
| Frequency of dosage of ART drug | | | | |
| Once | 31 (52.5%) | 144 (44.9%) | 175 (46.1%) | 1.184, 1 |
| Twice | 28 (47.5%) | 177 (55.1%) | 205 (53.9%) | |
| Taking medication other than ART | | | | |
| Yes | 41 (69.5%) | 126 (39.3%) | 167 (43.9%) | 18.501, 1 |
| No | 18 (30.5%) | 195 (60.7%) | 213 (56.1%) | |
| Presence of ART drug side effect | | | | |
| Yes | 8 (13.6%) | 11 (3.4%) | 19 (5.0%) | 10.772, 1 |
| No | 51 (86.4%) | 310 (96.6%) | 361 (95.0%) | |
| Current CD4 count | | | | |
| <200 cells/mm$^3$ | 5 (8.5%) | 29 (9%) | 34 (8.9%) | 4.199, 3 |
| 200–350 cells/mm$^3$ | 18 (30.5%) | 83 (25.9%) | 101 (26.6%) | |
| 351–500 cells/mm$^3$ | 22 (37.3%) | 91 (28.3%) | 113 (29.7%) | |
| >500 cells/mm$^3$ | 14 (23.7%) | 118 (36.8%) | 132 (34.7%) | |
| Current WHO stage | | | | |
| I | 54 (91.5%) | 309 (96.3%) | 363 (95.5%) | 2.616, 1 |
| ≥II | 5 (8.5%) | 12 (3.7%) | 17 (4.5%) | |
| Presence of opportunistic infections | | | | |
| Yes | 5 (8.5%) | 10 (3.1%) | 15 (3.9%) | 3.776, 1 |
| No | 54 (91.5%) | 311 (96.9%) | 365 (96.1%) | |
| HIV serostatus disclosure to any person | | | | |
| Yes | 44 (74.6%) | 301 (93.8%) | 345 (90.8%) | 21.956, 1 |
| No | 15 (25.4%) | 20 (6.2%) | 35 (9.2%) | |
| Currently drink alcohol | | | | |
| Yes | 13 (22%) | 65 (20.2%) | 78 (20.5%) | 0.097, 1 |
| No | 46 (78%) | 256 (79.8%) | 302 (79.5%) | |
| Currently chew chat | | | | |
| Yes | 5 (8.5%) | 21 (6.5%) | 26 (6.8%) | 0.292, 1 |
| No | 54 (91.5%) | 300 (93.5%) | 354 (93.2%) | |
| Currently smoke cigarettes | | | | |
| Yes | 1 (1.7%) | 8 (2.5%) | 9 (2.4%) | 0.137, 1 |
| No | 58 (98.3%) | 313 (97.5%) | 371 (97.6%) | |
| Perceived stigma | | | | |
| Yes | 42 (71.2%) | 72 (22.4%) | 114 (30.0%) | 56.418, 1 |
| No | 17 (28.8%) | 249 (77.6%) | 266 (70.0%) | |

## Factors associated with depression

In this study, a p-value of 0.2 was used to screen eligible variables for multivariable logistic regression analysis. As a result, 13 out of 22 variables (age, sex, ethnicity, level of education, occupation, living condition, wealth index, taking medication other than ART, drug side effect, perceived stigma, presence of opportunistic infection, and HIV serostatus disclosure) became eligible for multivariable logistic regression analysis. In the multivariable logistic regression

analysis, age, perceived stigma, taking medication other than ART and history of opportunistic infections showed a statistically significant association with depressive symptoms.

Patients aged 40–49 years were 2.9 times more likely to have depressive symptoms compared to patients aged 30–39 years (AOR = 2.96, 95% CI: (1.01, 8.68)). Similarly, the odds of depressive symptoms among patients aged greater than or equal to 50 years was 3.8 times higher compared to patients aged 30–39 years (AOR = 3.81, 95% CI: (1.05, 13.8)).

HIV-positive patients who took medication other than ART were 2.6 times more likely to develop depressive symptoms as compared to patients who didn't take medication other than ART (AOR = 2.58, 95% CI: (1.25, 5.33)). Similarly, patients who experienced perceived stigma were 10.2 times more likely to have depressive symptoms as compared to patients who didn't experience perceived stigma (AOR = 10.2, 95%CI: (4.26, 24.4)). The odds of depressive symptoms among patients who had a history of opportunistic infections was 5.2 times higher compared to their counterparts (AOR = 5.17, 95% CI: (1.31, 20.4)) (Table 3).

## Discussion

This study assessed the prevalence of depressive symptoms and associated factors among HIV-positive patients in Dessie town. The prevalence of depressive symptoms among HIV-positive patients was 59 (15.5%) with a 95% CI: (12.4%, 19.2%). Age, perceived stigma, taking medication other than ART and history of opportunistic infections were factors associated with depressive symptoms.

The prevalence of depressive symptoms is in line with a study done in Aksum town (14.6%) [9]. It is lower than studies conducted in Brazil (57.7%) [31], Bihar, India (57%) [32], Pradesh, India (67.3%) [33], Jos, Nigeria (31%) [29] and Southwest Nigeria (39.6%) [34]. The finding is also lower than studies conducted in Cameron (26.7%) [28], Sudan (63.1%) [35], Harar (45.8%) [11], Tigray (43.9%) [22], Alert Hospital (41.2%) [23] and Hawassa (48.6%) [10].

But the prevalence of depressive symptoms is slightly higher than a study conducted in Debre Markos town (11.7%) [8]. This discrepancy could be due to the difference in the study setting, study period, sample size and difference in the depression diagnostic tools used. For example, in our study, the PHQ-9 depression scale at a cutoff of 5 was used to assess depression. But, a study conducted in Pardesh, India used Montgomery Asberg Depression Rating Scale (MADRS) to assess depression [33]. Likewise, the Hamilton Depression Rating Scale (HAM-D) at a cut-off point of 7 and the Hospital Anxiety Depression Scale (HADS) cut-off points of 8 were used to assess depressive symptoms among studies done in Tigray and Alert hospital respectively [22, 23]. Studies conducted in Jos Nigeria, Cameron and Harar used the PHQ-9 depression scale at a cutoff of 5 to assess depression [11, 28, 29].

Being 40–49 years old and being ≥50 years old had a positive association with depressive symptoms. The finding is in line with a study conducted in Jos, Nigeria [29]. This might be because older age usually had an association with other chronic illnesses like diabetic Malthus, hypertension, heart disease, and kidney disease. A recent study showed that patients with comorbid illness had six times the risk for depressive symptoms than their counterparts [24]. In addition to this, chronic HIV disease and its challenges are more severe in older age patients due to low immune systems. This predisposes them to the vulnerability of depressive symptoms.

Having perceived stigma had a positive association with depressive symptoms. The finding is supported by studies conducted in Alert Hospital [23] and Debre Markos town [8]. It is also in line with studies conducted in Hawassa town [10] and Botswana [25]. HIV/AIDS is one of the chronic life-long diseases which is highly liable to stigma and discrimination. HIV-positive patients usually prefer to be alone to avoid stigma or discrimination. They also lose their work

**Table 3. Factors associated with depressive symptoms among HIV positive patients attending public health facilities of Dessie town, Northeast Ethiopia, 2019.**

| Variables | COR[1](95%CI[2]) | p-value | AOR[3](95%CI) | p-value |
|---|---|---|---|---|
| Age | | | | |
| 18–29 years | 3.34 (1.67, 6.70) | 0.005 | 1.93 (0.79, 4.70) | 0.145 |
| 30–39 years | 1 | | 1 | |
| 40–49 years | 1.49 (0.63, 3.53) | | **2.96 (1.01, 8.68)** | **0.048** |
| ≥50 years | 2.73 (1.006, 7.42) | | **3.81 (1.05, 13.8)** | **0.041** |
| Sex | | | | |
| Male | 0.64 (0.36, 1.14) | 0.131 | 0.68 (0.30, 1.55) | 0.365 |
| Female | 1 | | 1 | |
| Marital status | | | | |
| Single | 1.09 (0.63, 1.91) | 0.76 | | |
| Married | 1 | | | |
| Residence | | | | |
| Urban | 1 | | | |
| Rural | 1.07 (0.55, 2.10) | 0.841 | | |
| Ethnicity | | | | |
| Amhara | 1 | | 1 | |
| Others[a] | 0.32 (0.07, 1.36) | 0.122 | 0.34 (0.06, 2.02) | 0.235 |
| Religion | | | | |
| Christian | 1.09 (0.62, 1.90) | 0.774 | | |
| Muslim | 1 | | | |
| Level of education | | | | |
| Have no formal education | 1.57 (0.76, 3.27) | 0.066 | 1.88 (0.69, 5.16) | 0.220 |
| Grade1-8 | 1 | | 1 | |
| Grade 9–12 | 1.07 (0.49, 2.33) | | 0.74 (0.27, 2.05) | 0.566 |
| College and above | 2.78 (1.24, 6.24) | | 1.67 (0.38, 7.37) | 0.501 |
| Occupation | | | | |
| Government employed | 3.13 (1.37, 7.13) | 0.090 | 1.33 (0.31, 5.77) | 0.702 |
| Private employed | 2.03 (0.85, 4.84) | | 2.17 (0.76, 6.20) | 0.149 |
| Housewife | 2.18 (0.91, 5.20) | | 2.32 (0.72, 7.42) | 0.157 |
| Farmer | 1.70 (0.74, 3.93) | | 1.35 (0.44, 4.12) | 0.600 |
| Others[b] | 1 | | 1 | |
| Living condition | | | | |
| Family | 1 | | 1 | |
| Alone | 2.17 (1.07, 4.39) | 0.031 | 0.76 (0.21, 2.69) | 0.671 |
| Wealth index | | | | |
| Poorest quintile | 1.18 (0.53, 2.62) | 0.054 | 1.28 (0.47, 3.49) | 0.624 |
| Poorer quintile | 0.66 (0.29, 1.53) | | 0.65 (0.22, 1.93) | 0.434 |
| Middle quintile | 0.53 (0.18, 1.63) | | 0.66 (0.17, 2.56) | 0.546 |
| Richer quintile | 0.33 (0.13, 0.86) | | 0.37 (0.11, 1.27) | 0.115 |
| Richest quintile | 1 | | 1 | |
| Number of ART pills taken per day | | | | |
| 1 | 1 | | | |
| 2 | 0.68 (0.32, 1.46) | 0.559 | | |
| 3 | 0.78 (0.42, 1.46) | | | |
| Frequency of dosage of ART drug | | | | |
| Once | 1.36 (0.78, 2.37) | 0.278 | | |
| Twice | 1 | | | |

(*Continued*)

**Table 3.** (Continued)

| Variables | COR[1](95%CI[2]) | p-value | AOR[3](95%CI) | p-value |
|---|---|---|---|---|
| Taking medication other than ART | | | | |
| Yes | 3.53 (1.94, 6.41) | 0.000 | **2.58 (1.25, 5.33)** | **0.011** |
| No | 1 | | 1 | |
| Presence of ART drug side effect | | | | |
| Yes | 1 | | 1 | |
| No | 0.23 (0.09, 0.59) | 0.002 | 0.42 (0.13, 1.43) | 0.166 |
| History of opportunistic infections | | | | |
| Yes | 2.88 (0.95, 8.75) | 0.062 | **5.17 (1.31, 20.4)** | **0.019** |
| No | 1 | | 1 | |
| HIV sero-status disclosure[4] | | | | |
| Yes | 1 | | 1 | |
| No | 5.13 (2.45, 10.76) | 0.000 | 1.24 (0.35, 4.43) | 0.744 |
| Currently drink alcohol | | | | |
| Yes | 1.11 (0.57, 2.18) | 0.775 | | |
| No | 1 | | | |
| Currently chew chat | | | | |
| Yes | 1.32 (0.48, 3.66) | 0.590 | | |
| No | 1 | | | |
| Currently smoke cigarettes | | | | |
| Yes | 0.68 (0.08, 5.50) | 0.713 | | |
| No | 1 | | | |
| Perceived stigma | | | | |
| Yes | 8.54 (4.59, 15.91) | 0.000 | **10.2 (4.26, 24.4)** | **0.000** |
| No | 1 | | 1 | |

[1] crude odds ratio.

[2] confidence interval.

[3] adjusted odds ratio.

[4] the question used to define the variable "HIV serostatus disclosure" is have you ever disclosed your HIV serostatus disclosure to someone else like a sexual partner, families, friends, relatives, etc.

and social values, develop poor self-image and feel valueless as a result of stigma or discrimination. All of these factors contribute to the development of depressive symptoms [36].

The odds of depressive symptoms were 2.6 times more among patients who took medication other than ART as compared to their counterparts. In Ethiopia, HIV-positive patients are on efavirenz and nevirapine-based ART medications. Although ART medication can keep the patients healthy, it is associated with adverse drug reactions and drug toxicity [37]. The most common side effects associated with efavirenz and nevirapine-based ART medications are hepatotoxicity and central nervous system disorders typically, depression and sleep disorder [38, 39]. Taking other medication in addition to ART drugs may cause pill burden, worsen drug toxicity leading to poor medication adherence and depressive symptoms [13, 40, 41].

Patients who had a history of opportunistic infections were more likely to develop depressive symptoms as compared to those who had no history of opportunistic infections. The reason could be, patients with opportunistic infection might be excessively worried about their health and their physical appearance which might be a cause for the occurrence of depressive symptoms. The finding is supported by studies conducted in Fitche, Gimbi and Addis Ababa [12, 24, 42].

## Limitation of the study

This study includes patients who came to the health facility for drug refill. So, the finding cannot be generalized to all people living with HIV because it misses individuals with severe depressive symptoms who may not attend clinics regularly. This study is also prone to information bias which affects the prevalence of depressive symptoms. The issue of establishing a causal relationship due to the cross-sectional nature of the study is the other potential limitation of the study.

## Conclusions

The prevalence of depressive symptoms was low compared to the previous studies. Being 40–49 years old, being ≥50 years old, having perceived stigma, taking medication other than ART and history of opportunistic infections had a statistically significant association with depressive symptoms. Health care providers who are working in ART clinics should give special attention to older individuals, individuals who took medication other than ART, patients who experienced perceived stigma and patients with a history of opportunistic infections. Programs on counseling should be strengthened and health education should be given for the patients and the community at large to decrease HIV-related stigma. Further study with a qualitative study design would be recommended to support this finding.

## Supporting information

**S1 Table. Strengthening the Reporting of Observational Studies in Epidemiology (STROBE) 2007 (v4) Statement checklist for cross-sectional studies.**
(DOCX)

**S1 Dataset. The data set used to determine the prevalence of depression and associated factors among HIV-infected patients attending public health facilities of Dessie town.**
(SAV)

## Acknowledgments

The authors acknowledged Dessie city administrative health office and the respective study health facilities for their cooperation and data collectors, supervisors, and study subjects for their participation during the data collection period.

## Author Contributions

**Conceptualization:** Yitayish Damtie.

**Data curation:** Bezawit Adane.

**Formal analysis:** Yitayish Damtie, Mastewal Arefaynie, Atsedemariam Andualem.

**Investigation:** Yitayish Damtie, Afework Edmealem.

**Methodology:** Yitayish Damtie, Afework Edmealem.

**Software:** Yitayish Damtie, Melaku Yalew.

**Supervision:** Bezawit Adane.

**Validation:** Bereket Kefale.

**Writing – original draft:** Yitayish Damtie, Bereket Kefale.

**Writing – review & editing:** Yitayish Damtie, Melaku Yalew, Mastewal Arefaynie, Atsedemariam Andualem.

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
