## [Decision Letter · Decision Letter 0]

19 Apr 2021

PONE-D-21-03982

Depression and associated factors among HIV infected patients attending Public Health Facilities of Dessie town: A cross-sectional study.

PLOS ONE

Dear Dr. Damtie,

Thank you for submitting your manuscript to PLOS ONE. After careful consideration, we feel that it has merit but does not fully meet PLOS ONE’s publication criteria as it currently stands. Therefore, we invite you to submit a revised version of the manuscript that addresses the points raised during the review process.

We look forward to receiving your revised manuscript.

Kind regards,

Colette Smith, PhD

Academic Editor

PLOS ONE

Additional Editor Comments:

Please carefully consider the comments from the reviewers in your revision, particularly regarding making sure the language is clear and that it is clear that this is a cross-sectional study which cannot investigate cause-and-effect, and so you can only investigate associations rather than predictors.

Journal Requirements:

Reviewers' comments:

Reviewer's Responses to Questions

**Comments to the Author**

1. Is the manuscript technically sound, and do the data support the conclusions?

Reviewer #1: Partly

Reviewer #2: Yes

2. Has the statistical analysis been performed appropriately and rigorously? 

Reviewer #1: Yes

Reviewer #2: Yes

3. Have the authors made all data underlying the findings in their manuscript fully available?

Reviewer #1: Yes

Reviewer #2: No

4. Is the manuscript presented in an intelligible fashion and written in standard English?

Reviewer #1: Yes

Reviewer #2: No

5. Review Comments to the Author

Reviewer #1: This manuscript describes the results of a cross-sectional survey during one month in 2019 in several hospitals and health centres in Dessie Town in Ethiopia, to assess the prevalence of depression and factors associated with depression among persons living with HIV. Participants had to be linked into HIV care at one of the health centres and had to have been on ART for at least 6 months. A wide range of data were collected from face-to-face interviews with study nurses, and with further data extracted from clinical records. Assessing and thereby highlighting the extent of depression among PLWH is a good objective. There are a number of ways in which the manuscript should be revised to exploit the data collected more fully.

Introduction:

1) Line 54 (and also Abstract line 25): The statement that depression affects 121 million HIV infected patients worldwide is not correct (given that there are currently 37.9 million PLWH globally) – indeed reference 5 quotes this figure as the global total number of people with depression in 2012.

2) As this is a cross-sectional study, the objective would be to determine the prevalence of depression and other factors. Here and elsewhere (line 78, 153-154, 170, 173 etc.) please avoid the term ‘magnitude’ (instead of prevalence) at his term might be understood to refer to the severity of depression.

3) Please cite the meta-analysis (line 55–56) and cite some more recent reviews on depression, stigma and associated factors to put the study in context.

Methods:

Please provide more precise details on the methods, including:

1) Please provide more details of the systematic random sampling (not systematic sampling) approach and how this was divided between the two hospitals and three health centres. How were patients were invited to take part in the study?

2) It is important to know more of how the sampling was done so as to assess generalisability and bias in the selection of study participants. Restricting to individuals linked into care and on ART may exclude individuals with severe depression who may not attend clinic regularly.

3) The sample size calculation is difficult to follow: Is the assumption that 89% of PLWH will have depression? How do drug side effects, a 1:1 ratio of exposed to unexposed (which exposure?) affect the calculation, and what does the assumed odds ratio of 4.7 refer to?

4) I assume that 388 individuals were invited to take part, and 380 agreed? This could be more explicit. The 380 figure is more relevant for the abstract.

5) The PHQ-9 was used to assess for depression, using the cut-point of 5 to include any depression (including mild depression). Did the authors look at severe depression as well?

6) Since perceived stigma is a main outcome of the analysis, please specify how this was measured (more detail on the 10-question instrument) and how it was scored (range of responses, overall cut points etc). How does it work to use the median as the cut-point?

7) Statistical analysis – what do you mean by “using texts”?

8) How does the standard error predict multicollinearity?

9) Please clarify other variables / associated factors, specifically

a. Meal frequency – is this a socio-economic measure of how many meals/day the person can afford, or an indication of dietary habits?

b. How is the wealth index calculated, i.e. what factors is it based on, and has it been validated? I assume the strata are presented as quintiles, so should be referred to as ‘the richest quintile’ (line 139).

c. If meal frequency is a socio-economic measure, there may be collinearity with the wealth index.

d. the clinical, psycho-social and behavioural variables also need clearer definitions. E.g. please clarify the ‘number of tablets taken per day’ and ‘frequency of dosage’ variables – does this refer to the ARV medication, other medications or the sum of pill burden, and what does a dosage of ‘once’ and ‘twice’ refer to?

e. How is the ‘side effects’ variable defined, which side effects are of interest, and from which treatment or medication (ARVs or other medication)?

f. Please list the conditions included in ‘comorbid illnesses’ and ‘opportunistic infections’ – are you referring to non-AIDS and AIDS-related clinical conditions? How do these variables relate to ‘concomitant illness’ reported in Table 3?

g. Please explain the ‘HIV disclosure’ variable vs. who reported disclosing what to whom?

h. Please specify for alcohol and Chat use and for smoking status, are you looking at current or ever use?

10) Line 120: Why were the variables considered predictors of ART adherence?

1) Data presentation: Tables 1 and 2 would be more informative if they showed data stratified by the outcome, depression (i.e. present numbers and % of those with depression and those without, in addition to the overall totals) with a chi-squared test for significance. Stratified data should be shown for all variables, not only those selected in Table 3.

2) The descriptions for Table 1 (lines 134-139) and Table 2 (lines 144-149) simply reiterate apparently random lines from the tables. If the tables were stratified according to whether participants had depression or not, this would allow a description of how depression varies across the different variables.

3) Please state which variables were included in the multivariable regression models? A table showing the unadjusted odds ratios for all variables, and the adjusted odds ratios for those included in the multivariable model would be more informative. P values should be listed in full, not categorised to *.

4) How were the reference categories chosen, e.g. for age or education. For age, the category of ≥50 years, which contains fewer than 10% of the individuals overall is shown as the reference category. If, alternatively, the largest age group, 30-39 is used as reference, individuals in the youngest age category (18-29 years old) have more than 3x the odds of depression than those 30-39 years old (unadjusted OR 3.34, CI 1.67-6.70, = 0.0007), which would agree with the high rates of depression observed in adolescents and adults under 25 years old in other populations.

5) While it is not clear how these variables are defined, could there be collinearity between ‘Taking medication other than ART’ and ‘concomitant illness’.

6) The main factors associated that are commented on in the paper are perceived stigma, taking medication other than ART, and living alone. But other variables also showed significant associations (stage of AIDS, alcohol use and drug side effects, and younger age if the reference group is changed as I indicated). Why are these other factors not described?

Discussion

1) Since this is a cross-sectional study, cause and effect are less clear, and the variables found to be associated with depression should not be described as predictors (line 172, also line 42). Indeed, depression might lead someone to prefer to live alone, or to use alcohol.

2) Please describe possible reasons for the observed associations in more detail, and how the associated factors might be related, e.g. have you considered whether there is collinearity for individuals to suffer from other illnesses and using medications other than ARVs.

3) Please describe the strengths and limitations of the study more fully, including selection bias and information bias and how these may affect the study results, and generalisability to PLWH in Ethiopia and elsewhere.

4) Lines 180-185: Listing numbers of diagnosed individuals in the different comparator studies is meaningless without knowing denominators and does not inform the discussion

Minor errors:

There are some grammatical errors or omission of words; these are too numerous to list all, but some examples are:

- Line 27: HIV infected patients attending or HIV infected patients who attended.

- As per UN terminology guidelines, avoid the terms ‘HIV-infected’ or ‘patient’ in a setting such as this – these descriptors may themselves be stigmatising.(https://www.iasociety.org/Web/WebContent/File/unaids_terminology_guide_en.pdf)

- Line 43-44 add the word 'who': patients who experienced (persons who experienced)

- Remove the extra space in Line 49 - the figure should be 399,000

- Line 70: CD4 – the 4 is not a subscript (also elsewhere)

- Line 71 change 'has' to 'have' in factors linked to depression has been recognized

- Line 85 change 'adults who attending' to 'adults who attended'

- Line 98 missing word 'was' in The patient's card also reviewed…

- Line 100 missing word 'was' in The questionnaire composed of

- Lines 134-139 – please ensure consistent use of the past tense

- Line 161 should be the odds of depression

- Line 167 – what is meant by bi-variable? These are univariable/unadjusted/crude odds ratios with one dependent and one independent variable.

Data would be fully available on request

Reviewer #2: I think this is an interesting paper that adds to the literature. However, it is not well written and needs extensive editing.

1) In the abstract the first sentence is confusing. Is the 121 million people those with HIV or HIV and depression? You need to put how depression was diagnosed in the abstract- both the tool used and the cutoff used.

2) Page 3, line 53- most common not commonest

3) Page 3 lines 55-57 you need a citation. There has been a review on the prevalence of depression in Africa and a CDC survey of multiple African countries. Lofgren, S.M., al. Burden of Depression in Outpatient HIV-Infected adults in Sub-Saharan Africa; Systematic Review and Meta-analysis. AIDS Behav 24, 1752–1764 (2020). https://doi.org/10.1007/s10461-019-02706-2

Seth P, Kidder D, Pals S, et al. Psychosocial functioning and depressive symptoms among HIV-positive persons receiving care and treatment in Kenya, Namibia, and Tanzania. Prev Sci. 2014;15(3):318–28.

4) There are many sentences that are long and compound. These should be split. Examples Page 3 line 60-64, Page 3 67-71. There are also multiple paragraphs with one sentence. Paragraphs have at least 3. Examples Page 3 line 67-71, Page 4 72-75.

5) The sample size calculation on pages 4-5 is confusing to me. I don't understand what the 89.9% depression means.

6) Results page 6- The pulling out of the percentage of age 18-29 was odd. Why not use median age or present multiple categories. The spacing in this paragraph has multiple errors. Finally the sentences included multiple unrelated datapoints. Ex people live with their families, eat 3+ meals, % of richest. The % eating 3+ meals and wealth category are related but living with one's family is note.

7) Page 9 you reference tables and I assume you mean ART pill but you need to say that. Again, you have many compound complex sentences that need to be broken up.

8) again in the discussion there are paragraphs with 2 sentences. The sentences are not really arranged into coherent paragraphs.

9) You need a more nuanced comparison between studies. You start it but the tool used and the cutoff are critical in comparing incidence and prevalence. Also if the individuals were just diagnosed with HIV. Or if they were otherwise sick such as with TB. Finally are they inpatient sor outpatients. All of these factors need a more careful discussion.

10) You use the cutoff of 5 with the PHQ-9 which is a less common cutoff to use. It is the cutoff for mild depression. The cutoff of 10 is more commonly used. I think you can use it if you justify what it is used but more justification is needed. Also you should discuss the cutoff when you compare to other studies.

11) In the discussion you reference ART side effects and toxicity as possible causes of depression. However, you do not state what ART the individuals are on. Most of Africa switched to dolutegravir based therapy in 2019. You can make the claim of side effects if your participants were on efavirenz but most individuals on dolutegravir have almost no side effects.

12) You have a limitations paragraph before the conclusion but this should be labeled as such.

13) The first sentence of the conclusion is really confusing and should be re-written.

14) Throughout paper lots of grammatical errors

6. PLOS authors have the option to publish the peer review history of their article (what does this mean?). If published, this will include your full peer review and any attached files.

Reviewer #1: No

Reviewer #2: No

---

## [Author Response · Author response to Decision Letter 0]

3 Jul 2021

Colette Smith, PhD

RE: Manuscript ID: PONE-D-21-03982 (Depression and associated factors among HIV infected patients attending Public Health Facilities of Dessie town: A cross-sectional study.)

Dear Dr. Colette Smith,

Thank you very much for your email and the comments/suggestions of the reviewers and academic editor. We have looked at the comments and have revised our paper accordingly. We hope our paper improved as a result of incorporating the reviewers' and academic editor's comments and suggestions.

Please find for your kind consideration the following:

A rebuttal letter that responds to each point raised by the academic editor and reviewer labeled 'Response to Reviewers'. The point-by-point responses are written in italic font style.

A revised manuscript with track changes labeled 'Revised Manuscript with Track Changes'.

A revised paper without tracked changes labeled 'Manuscript'

While hoping that these changes would meet with your favorable consideration, we are happy to hear if there are more comments and suggestions. Please do not hesitate to let us know if you have any questions.

Yours Sincerely,

Yitayish Damtie 

School of Public Health, Wollo University 

Dessie, Ethiopia

Tel:+251943517982

E-mail: yitutile@gmail.com

Point by point response

Additional Editor Comments:

Please carefully consider the comments from the reviewers in your revision, particularly regarding making sure the language is clear and that it is clear that this is a cross-sectional study that cannot investigate cause-and-effect, and so you can only investigate associations rather than predictors.

Thank you. We have tried to make sure the language clear and revise our manuscript based on reviewers' comments accordingly. In addition, we have tried to use the term associated factors instead of predictors throughout the manuscript accordingly. 

Journal Requirements:

1. Please ensure that your manuscript meets PLOS ONE's style requirements, including those for file naming. The PLOS ONE style templates can be found at:

https://journals.plos.org/plosone/s/file?id=wjVg/PLOSOne_formatting_sample_main_body.pdfAndhttps://journals.plos.org/plosone/s/file?id=ba62/PLOSOne_formatting_sample_title_authors_affiliations.pdf

Thank you. We have tried to organize our manuscript based on PLOS ONE's style requirements including those for file naming.

a) We ask great excuse for this. There are no legal or ethical restrictions on sharing data publicly. 

A) If there are ethical or legal restrictions on sharing a de-identified data set, please explain them in detail (e.g., data contain potentially identifying or sensitive patient information) and who has imposed them (e.g., an ethics committee). Please also provide contact information for a data access committee, ethics committee, or other institutional body to which data requests may be sent.

Thank you for your comment. But there are no ethical or legal restrictions on sharing the data set of this study publicly. 

We have tried to upload the data set of this study as a supporting information file accordingly.

Reviewer #1: 

This manuscript describes the results of a cross-sectional survey during one month in 2019 in several hospitals and health centers in Dessie Town in Ethiopia, to assess the prevalence of depression and factors associated with depression among persons living with HIV. Participants had to be linked into HIV care at one of the health centers and had to have been on ART for at least 6 months. A wide range of data were collected from face-to-face interviews with study nurses, and with further data extracted from clinical records. Assessing and thereby highlighting the extent of depression among PLWH is a good objective. There are several ways in which the manuscript should be revised to exploit the data collected more fully.

Thank you. We have tried to revise our manuscript accordingly. 

Introduction:

1) Line 54 (and also Abstract line 25): The statement that depression affects 121 million HIV infected patients worldwide is not correct (given that there are currently 37.9 million PLWH globally) – indeed reference 5 quotes this figure as the global total number of people with depression in 2012.

Yes, we acknowledge the problem. We have tried to delete it and revise the abstract and introduction section accordingly. 

2) As this is a cross-sectional study, the objective would be to determine the prevalence of depression and other factors. Here and elsewhere (line 78, 153-154, 170, 173 etc.) please avoid the term ‘magnitude’ (instead of prevalence) at his term might be understood to refer to the severity of depression.

Thank you. We have tried to use prevalence instead of magnitude throughout our manuscript (line 27, 42, 83, 168, 191, 195, 200 and 241

3) Please cite the meta-analysis (line 55–56) and cite some more recent reviews on depression, stigma and associated factors to put the study in context.

We have tried to cite the meta-analysis and more recent reviews in the introduction line 56-57 and line 59-60 accordingly. 

Methods: 

Please provide more precise details on the methods, including:

1) Please provide more details of the systematic random sampling (not systematic sampling) approach and how this was divided between the two hospitals and three health centers. How patients were invited to take part in the study?

We have tried to provide a detail description regarding systematic random sampling and its approach in the method section line 99-107. 

2) It is important to know more of how the sampling was done to assess generalisability and bias in the selection of study participants. Restricting to individuals linked into care and on ART may exclude individuals with severe depression who may not attend clinic regularly.

Yes, you are right. The study was restricted to individuals linked into care and on ART. This may underestimate depression because it is prone to miss individuals with severe depression who may not attend the clinic regularly. Since it is one of the potential limitations of this study, we have tried to include it under the limitation section line 234-236. 

3) The sample size calculation is difficult to follow: Is the assumption that 89% of PLWH will have depression? How do drug side effects, a 1:1 ratio of exposed to unexposed (which exposure?) affect the calculation, and what does the assumed odds ratio of 4.7 refer to?

Thank you for your comment. We used epi Info software to calculate the sample size. To calculate sample size using Epi Info, one requires to provide the following information i.e. confidence level: usually set at 95%, power: usually set at 80%, % of outcome in unexposed group: the proportion of unexposed patients with the outcome of interest (in our case, the proportion of depression among HIV patients without ART drug side effect), the ratio of exposed to unexposed: usually taken as 1 for cross-sectional study and adjusted odds ratio of the exposure variable taken from the previous studies. We have tested several exposure variables which are taken from previous literature to get the maximum sample size. As a result ART drug side effects from Beyene Gebrezgiabher B, et al (reference 12) gives the maximum sample size. So, 89% is the proportion of depression among HIV patients who don’t experience ART drug side effects and 4.7 is the AOR of ART drug side effects. 

4) I assume that 388 individuals were invited to take part, and 380 agreed? This could be more explicit. The 380 figure is more relevant for the abstract.

Yes, 388 individuals were invited to take part, and 380 agreed. We have tried to make it 380 in the abstract section line 29. 

5) The PHQ-9 was used to assess for depression, using the cut-point of 5 to include any depression (including mild depression). Did the authors look at severe depression as well?

Yes. 

6) Since perceived stigma is a main outcome of the analysis, please specify how this was measured (more detail on the 10-question instrument) and how it was scored (range of responses, overall cut points etc). How does it work to use the median as the cut-point?

Thank you. As we have stated in the manuscript, perceived stigma was measured using ten questions with ‘yes’ or ‘no’ response options. These questions were adapted from the HIV stigma index validation survey conducted in six Iranian cities. It has been validated in our country as well (in Iimma town) and hasn't a cutoff point like the PHQ-9 depression scale. We calculate the overall score of perceived stigma out of ten and have checked its distribution. Since the data was not normally distributed, we use the median value to classify perceived stigmas as ‘yes’ or ‘no’. 

7) Statistical analysis – what do you mean by “using texts”?

It is to mean in text form (in narration). But since it is irrelevant we have deleted it. 

8) How does the standard error predict multicollinearity?

Thank you. As we know multicollinearity is a condition in which two or more predictors are highly correlated with one another. One of the features of multicollinearity is that the standard errors of the affected coefficients tend to be large. If the standard error is between -2 and + 2, we can say that there is no multicollinearity.

9) Please clarify other variables / associated factors, specifically

a. Meal frequency – is this a socio-economic measure of how many meals/day the person can afford, or an indication of dietary habits?

Yes. It is the socio-economic measure. In our country, the majority of the people afford three meals/day 

b. How is the wealth index calculated, i.e. what factors is it based on, and has it been validated? I assume the strata are presented as quintiles, so should be referred to as ‘the richest quintile’ (line 139).

Thank you. The wealth index was calculated for urban and rural areas separately using principal component analysis. It includes a wide range of factors like source of drinking water, type of toilet, owner of the house, number of class in the house, source of energy for food cooking, having mobile, cycle, motorcycle, car, ox bees, sheep gout, hen, etc. it has been validated and the strata were presented as quintiles. So, we have tried to revise it accordingly. 

c. If meal frequency is a socio-economic measure, there may be collinearity with the wealth index.

Thank you. We remove meal frequency from the analysis due to its collinearity with the wealth index.

d. the clinical, psycho-social and behavioral variables also need clearer definitions. E.g. please clarify the 'number of tablets taken per' day and 'frequency of dosage' variables – does this refer to the ARV medication, other medications or the sum of pill burden, and what does a dosage of 'once' and 'twice' refer to?

Thank you. ‘Number of tablets taken per day’ and ‘frequency of dosage’ refer to the ARV medication. When we say a dosage of ‘once’, the number of tablets might be either one or more than one but it is taken once per day (in medical terms it is commonly known as stat). On the other hand, when we say a dosage of ‘twice’, the tablet is taken twice per day regardless of its number (12 hours apart or commonly known as BID).

e. How is the ‘side effects’ variable defined, which side effects are of interest, and from which treatment or medication (ARVs or other medication)?

Thank you. The variable 'side effects' represents the side effect of ARV medication. It includes skin rash, anemia, gastrointestinal disorder, jaundice, sleep disorder and headache.

f. Please list the conditions included in ‘comorbid illnesses’ and ‘opportunistic infections’ – are you referring to non-AIDS and AIDS-related clinical conditions? How do these variables relate to ‘concomitant illnesses reported in Table 3?

Thank you. ‘Comorbid illnesses’ refers to non-AIDS illness. It includes hypertension, asthma, diabetic Malthus and cancer. On the other hand ‘opportunistic infections’ refers to AIDS-related clinical conditions like tuberculosis, herpes zoster, oral thrush, esophageal candidacies, pneumocystis carnie pneumonia etc. Concomitant illness is used to refer to comorbid illnesses. 

g. Please explain the ‘HIV disclosure’ variable vs. who reported disclosing what to whom?

Thank you. HIV serostatus disclosure was reported if study participants tell their serostatus to any other person i.e. to their sexual partner, family members, friends or relatives. 

h. Please specify for alcohol and Chat use and for smoking status, are you looking at current or ever use?

Thank you. We have to look at the current use. We have tried to specify it in Table 2 and 3

10) Line 120: Why were the variables considered predictors of ART adherence?

Thank you. It is an editorial issue, we accept the problem and correct it accordingly. 

1) Data presentation: Tables 1 and 2 would be more informative if they showed data stratified by the outcome, depression (i.e. present numbers and % of those with depression and those without, in addition to the overall totals) with a chi-squared test for significance. Stratified data should be shown for all variables, not only those selected in Table 3.

Thank you. We have tried to present the data stratified by the outcome, depression with a chi-squared test for significance in Table 1 and 2. But we delete the stratification presented in Table 3 because of repetition. 

2) The descriptions for Table 1 (lines 134-139) and Table 2 (lines 144-149) simply reiterate apparently random lines from the tables. If the tables were stratified according to whether participants had depression or not, this would allow a description of how depression varies across the different variables.

We have tried to stratified the tables with respect to the outcome variable and rewrite the descriptions for Table 1 and 2 in the result section line 149-155 and line 160-164 respectively. 

3) Please state which variables were included in the multivariable regression models? A table showing the unadjusted odds ratios for all variables, and the adjusted odds ratios for those included in the multivariable model would be more informative. P values should be listed in full, not categorised to *.

We have tried to list variables included in the multivariable regression models in the result section line 173-176. We also put the unadjusted odds ratios for all variables, the adjusted odds ratios for those included in the multivariable model, and the p-value in Table 3 of the revised manuscript. 

4) How were the reference categories chosen, e.g. for age or education. For age, the category of ≥50 years, which contains fewer than 10% of the individuals overall is shown as the reference category. If, alternatively, the largest age group, 30-39 is used as reference, individuals in the youngest age category (18-29 years old) have more than 3x the odds of depression than those 30-39 years old (unadjusted OR 3.34, CI 1.67-6.70, = 0.0007), which would agree with the high rates of depression observed in adolescents and adults under 25 years old in other populations.

Thank you. We have tried to change the reference category for both variable in Table 3 accordingly. 

5) While it is not clear how these variables are defined, could there be collinearity between ‘Taking medication other than ART’ and ‘concomitant illness’.

Thank you. We removed concomitant illness from the analysis due to its colinearity with taking medication other than ART

6) The main factors associated that are commented on in the paper are perceived stigma, taking medication other than ART, and living alone. But other variables also showed significant associations (stage of AIDS, alcohol use and drug side effects, and younger age if the reference group is changed as I indicated). Why are these other factors not described?

Thank you. We have changed the reference group for these variables (stage of AIDS, alcohol use and drug side effects, and younger age) accordingly. But none of them showed significant associations except age. 

Discussion

1) Since this is a cross-sectional study, cause and effect are less clear, and the variables found to be associated with depression should not be described as predictors (line 172, also line 42). Indeed, depression might lead someone to prefer to live alone or to use alcohol.

Thank you. We have tried to make it associated factors in the abstract and discussion section line 41 and 195 respectively. 

2) Please describe possible reasons for the observed associations in more detail, and how the associated factors might be related, e.g. have you considered whether there is collinearity for individuals to suffer from other illnesses and using medications other than ARVs.

We have tried to make the detail of the reasons and removed the variable ‘concomitant illness’ from the analysis. 

3) Please describe the strengths and limitations of the study more fully, including selection bias and information bias, and how these may affect the study results, and generalizability to PLWH in Ethiopia and elsewhere.

Thank you for your comment. Discussing the strengths of this study is not as important since our study shared all the strengths of a certain cross-sectional study. However, we have tried to describe the limitations of the study in detail (line 235-240). 

4) Lines 180-185: Listing numbers of diagnosed individuals in the different comparator studies is meaningless without knowing denominators and does not inform the discussion

Thank you. We have tried to delete it from paragraph three of the discussion section accordingly. 

Minor errors:

There are some grammatical errors or omission of words; these are too numerous to list all, but some examples are:

Thank you. We have tried to revise the grammatical errors accordingly. 

- Line 27: HIV infected patients attending or HIV infected patients who attended.

- As per UN terminology guidelines, avoid the terms 'HIV-infected or 'patient' in a setting such as this –these descriptors may themselves be stigmatizing. (https://www.iasociety.org/Web/WebContent/File/unaids_terminology_guide_en.pdf)

- Line 43-44 add the word 'who': patients who experienced (persons who experienced)

- Remove the extra space in Line 49 - the figure should be 399,000

- Line 70: CD4 – the 4 is not a subscript (also elsewhere)

- Line 71 change 'has' to 'have' in factors linked to depression has been recognized

- Line 85 change 'adults who attending' to 'adults who attended'

- Line 98 missing word 'was' in the patient's card also reviewed…

- Line 100 missing word 'was' in the questionnaire composed of

- Lines 134-139 – please ensure consistent use of the past tense

- Line 161 should be the odds of depression

- Line 167 – what is meant by bi-variable? These are univariable/unadjusted/crude odds ratios 

with one dependent and one independent variable.

Data would be fully available on request

Reviewer #2: 

I think this is an interesting paper that adds to the literature. However, it is not well written and needs extensive editing.

Thank you. We have tried to revise the whole part of the manuscript accordingly. 

1) In the abstract the first sentence is confusing. Is the 121 million people those with HIV or HIV and depression? You need to put how depression was diagnosed in the abstract- both the tool used and the cutoff used.

Thank you. We have tried to modify it accordingly. We put the tool and the cutoff used to assess depression in the abstract section line 32. 

2) Page 3, line 53- most common not commonest

Thank you. The comment is accepted and addressed accordingly. 

3) Page 3 lines 55-57 you need a citation. There has been a review on the prevalence of depression in Africa and a CDC survey of multiple African countries. Lofgren, S.M., al. Burden of Depression in Outpatient HIV-Infected adults in Sub-Saharan Africa; Systematic Review and Meta-analysis. AIDS Behav 24, 1752–1764 (2020). https://doi.org/10.1007/s10461-019-02706-2

Seth P, Kidder D, Pals S, et al. Psychosocial functioning and depressive symptoms among HIV-positive persons receiving care and treatment in Kenya, Namibia, and Tanzania. Prev Sci. 2014; 15(3):318–28.

Thank you. We have tried to put the citation for the meta-analysis and add additional literature in the introduction section line 56-57 and line 59-60 respectively.

4) There are many sentences that are long and compound. These should be split. Examples Page 3 line 60-64, Page 3 67-71. There are also multiple paragraphs with one sentence. Paragraphs have at least 3. Examples Page 3 line 67-71, Page 4 72-75.

Thank you. We acknowledged the problem have tried to split it accordingly.

5) The sample size calculation on pages 4-5 is confusing to me. I don't understand what the 89.9% depression means.

Thank you for your comment. We used epi Info software to calculate the sample size. To calculate sample size using Epi Info, one requires to provide the following information i.e. confidence level: usually set at 95%, power: usually set at 80% and % of outcome in unexposed group: the proportion of unexposed patients with the outcome of interest (in our case, the proportion of depression among HIV patients without ART drug side effect). We have tested several exposure variables which are taken from previous literature to get the maximum sample size. As a result ART drug side effects from Beyene Gebrezgiabher B, et al (reference 12) gives the maximum sample size. So, 89% is the proportion of depression among HIV patients without ART drug side effects and 4.7 is the AOR of ART drug side effects. 

6) Results page 6- The pulling out of the percentage of age 18-29 was odd. Why not use median age or present multiple categories. The spacing in this paragraph has multiple errors. Finally the sentences included multiple unrelated data points. Ex people live with their families, eat 3+ meals, % of richest. The % eating 3+ meals and wealth category are related but living with one's family is note.

Thank you. We have tried to revise it accordingly.

7) Page 9 you reference tables and I assume you mean ART pill but you need to say that. Again, you have many compound complex sentences that need to be broken up.

Thank you, the comment is accepted and addressed accordingly. 

8) again in the discussion there are paragraphs with 2 sentences. The sentences are not really arranged into coherent paragraphs.

We acknowledge the problem and tried to revise it accordingly.

9) You need a more nuanced comparison between studies. You start it but the tool used and the cutoff are critical in comparing incidence and prevalence. Also if the individuals were just diagnosed with HIV. Or if they were otherwise sick such as with TB. Finally are they inpatient or outpatients. All of these factors need a more careful discussion.

Thank you. We have tried to compare the tool used and the cutoff points in the discussion section line 204-210. But all the studies mentioned in the discussion section included HIV-infected outpatients in their study. So, there is no need to compare each study concerning the characteristics of their study participants.

10) You use the cutoff of 5 with the PHQ-9 which is a less common cutoff to use. It is the cutoff for mild depression. The cutoff of 10 is more commonly used. I think you can use it if you justify what it is used but more justification is needed. Also you should discuss the cutoff when you compare to other studies.

Thank you. But we left the cutoff of 5 as it is since the most published study used the cutoff of 5 to categorize patients as having depression and not having depression. 

11) In the discussion you reference ART side effects and toxicity as possible causes of depression. However, you do not state what ART the individuals are on. Most of Africa switched to dolutegravir based therapy in 2019. You can claim side effects if your participants were on efavirenz but most individuals on dolutegravir have almost no side effects.

Thank you. We have tried to state the type of ART drug that the individuals are on and its side effects in the discussion line 227-231. 

12) You have a limitations paragraph before the conclusion but this should be labeled as such.

The comment is accepted and addressed accordingly.

13) The first sentence of the conclusion is confusing and should be re-written.

Yes, you are correct. We have tried to re-write it accordingly. 

14) Throughout the paper lots of grammatical errors

Thank you. We have tried our best to revise our paper accordingly 

.

---

## [Editor Report · Decision Letter 1]

15 Jul 2021

PONE-D-21-03982R1

Depression and associated factors among HIV infected patients attending public health facilities of Dessie town: A cross-sectional study.

PLOS ONE

Dear Dr. Damtie,

Thank you for submitting your manuscript to PLOS ONE. After careful consideration, we feel that it has merit but does not fully meet PLOS ONE’s publication criteria as it currently stands. Therefore, we invite you to submit a revised version of the manuscript that addresses the points raised during the review process.

We look forward to receiving your revised manuscript.

Kind regards,

Colette Smith, PhD

Academic Editor

PLOS ONE

Journal Requirements:

Additional Editor Comments (if provided):

Thank you for revising and re-submitting your manuscript. I have the following suggestions for you to consider.

Throughout - please use "HIV positive" or "people living with HIV" or "PLWH" rather than "HIV-infected"

Throughout - please refer to "depressive symptoms" instead of "depression".

Abstract - Please clarify that Dessie town is in Ethiopia and give the calendar dates over which the survey was conducted

Abstract - please change line to "PHQ-9) at a cut-off point of 5 was used to assess depressive symptoms". This is because these tools screen for symptomology, rather than confer a clinical diagnosis.

Abstract, result: PLease change "The magnitude of depression among..." to "The prevalence of depression among..."

Abstract, result: Please add in what the comparison group was for the age association (e.g. "Age 40-49 compared to 30-39 years (AOR=3.00; "

Abstract, conclusion: Please change the first sentence to "The prevalence of depression was low compared to previous studies."

Introduction, line 52: Please reword as "In Ethiopia, it was estimated in 20XX that 399,000 of the urban population were living with HIV/AIDS."

Introduction, line 56: Please change to "It disproportionately affects people living with HIV, with a recent meta-analysis indicating a two-fold higher risk compared to the general population [8]".

Introduction, line 58-60 (sentence beginning "The lifetime prevalence of depression..."). I would suggest cutting this sentence as it contradicts the sentence in lines 55-56, and the context for these numbers is not clear.

Introduction, line 61-62. I would suggest rewording as "In Ethiopia, previous studies assessing the prevalence of depression among PLWH have differed considerably, with estimates ranging from 11.7-76.7% [11-15]."

Introduction, line 71. Please change to "Depression has been associated with a wide range of factors, including age, sex, ....".

Introduction, lines 78-81. Is this paragraph necessary? Although i completely agree that ending the HIV/AIDS epidemic is important, its direct link to depression requires more explanation, and is not directly relevant to the manuscript. So this section could be cut.

Methods, line 88. Please clarify that Dessie town is in Ethiopia, and perhaps indicate where Dessie is within Ethiopia (e.g. "A cross-sectional study was conducted between 1-30 March 2019 in public health facilities of Dessie Town, which is a city located within north-central Ethiopia").

Methods, line 91. SHould this be "The study population was randomly selected from all adults with HIV attending public health facilities...."

Methods, sample size. Do you mean that you assumed that 10.1% would have depressive symptoms (so that 89.9% did not)? Otherwise this prevalence seems a little high.

Methods, lines 95-108. This section is perhaps a little long. It could be considerable shortened. For example: "The study sample size was calculated for the primary risk factor as presence of antiretroviral drug side effects, considered as a binary variable. Based on previous research [12], we wished to to detect differences in adjusted odds ratios of at least 4.7, assuming an underlying prevalence of 10.1%, power of 80% and 5% Type I error. After adjustment for 10% non-response, the minimum required sample size was 388. Systematic random sampling was used to select participants using the antiretroviral drug dispenary registration book as the sampling frame. Study participants were invited to participate when attending the health facility for antiretroviral drug collection."

Methods, line 111. Please clarify if the card reviewed was the medical record?

Methods, line 114. Please consider rewording as "The questionnaire collected data on socio-demographic, clinical, psycho-social, and behavioural characteristics."

Methods, line 116. Please clarify that the PHQ-9 is use to measure depressive symptoms rather than depression directly.. Please also clarify that the cut-off of 5 typically corresponds to mild, moderate or severe depressive symptoms as other cut-offs are sometimes used

Methods, line 125. Please give the median score (out of 10?) used to define "high" and "low" stigma. Also, where was this median obtained from, as it appears that only 30% had perceived stigma in Table 2 (instead of the 50% you would expect if the median was taken from this present study).

Methods, line 131. as the study is cross-sectional, the independent variables are not really "predictors". Perhaps "explanatory variables", "factors" or "independent variables" would be better?

Methods, line 135. As this is a cross-sectional study, it may not be appropriate to use the word "determinents". "associated factors" could be an alternative?

Results, line 152. It would be good to give the overall prevalence of depression (for the whole population), before giving it in the sub-groups.

Results, Table 3. Would it be possible to add the question used to define the variable "HIV sero-status disclosure" in a footnote to the table?

Results, Table 3. CD4 count, opportunistic infections and WHO stage are quite correlated - is it reasonable to include all in the model?

Discussion, line 227. Please say "times the odds" rather than "times the risk" here.

Discussion, line 237. You could consider adding the word all, so it reads "cannot be generalised to all people living with HIV".

---

## [Author Response · Author response to Decision Letter 1]

20 Jul 2021

Colette Smith, PhD

RE-2: Manuscript ID: PONE-D-21-03982R1 (Depression and associated factors among HIV infected patients attending public health facilities of Dessie town: A cross-sectional study.)

Dear Dr. Colette Smith,

Thank you very much for your email and the comments/suggestions of the reviewers and academic editor. We have looked at the comments and have revised our paper accordingly. We hope our paper improved as a result of incorporating the reviewers' and academic editor's comments and suggestions.

Please find for your kind consideration the following:

A rebuttal letter that responds to each point raised by the academic editor and reviewer labeled 'Response to Reviewers'. The point-by-point responses are written in italic font style.

A revised manuscript with track changes labeled 'Revised Manuscript with Track Changes'.

A revised paper without tracked changes labeled 'Manuscript'

While hoping that these changes would meet with your favorable consideration, we are happy to hear if there are more comments and suggestions. Please do not hesitate to let us know if you have any questions.

Yours Sincerely,

Yitayish Damtie 

School of Public Health, Wollo University 

Dessie, Ethiopia

Tel:+251943517982

E-mail: yitutile@gmail.com

Point by point response

Journal Requirements:

Thank you. Reference #1, #18, #24, #29 and #30 have been retracted and replace with relevant current references. But reference #2 is retracted and used as it is due to the absence of relevant literature. We indicated its retracted status in the References list.

Additional Editor Comments (if provided):

Throughout - please use "HIV positive" or "people living with HIV" or "PLWH" rather than "HIV-infected"

Thank you. We have tried to use "HIV positive" throughout the document accordingly. 

Throughout - please refer to "depressive symptoms" instead of "depression".

Thank you. The comment is accepted and addressed accordingly. 

Abstract - Please clarify that Dessie town is in Ethiopia and give the calendar dates over which the survey was conducted.

Thank you. We have tried to clarify it in the abstract section line 29. 

Abstract - please change line to "PHQ-9) at a cut-off point of 5 was used to assess depressive symptoms". This is because these tools screen for symptomology, rather than confer a clinical diagnosis.

Thank you. The comment is accepted and addressed in the abstract section line 33-34. 

Abstract, result: PLease change "The magnitude of depression among..." to "The prevalence of depression among..."

We have tried to address it in the abstract section line 39.

Abstract, result: Please add in what the comparison group was for the age association (e.g. "Age 40-49 compared to 30-39 years (AOR=3.00; "

Thank you. The comment is accepted and addressed in the abstract section line 40-41 accordingly. 

Abstract, conclusion: Please change the first sentence to "The prevalence of depression was low compared to previous studies."

We have tried to address it in the abstract section line 45 accordingly

Introduction, line 52: Please reword as "In Ethiopia, it was estimated in 20XX that 399,000 of the urban population were living with HIV/AIDS."

Thank you. The comment is accepted and addressed in the introduction section line 55-56 accordingly. 

Introduction, line 56: Please change to "It disproportionately affects people living with HIV, with a recent meta-analysis indicating a two-fold higher risk compared to the general population [8]".

Thank you. We have addressed it in the introduction section line 58-59 accordingly

Introduction, line 58-60 (sentence beginning "The lifetime prevalence of depression..."). I would suggest cutting this sentence as it contradicts the sentence in lines 55-56, and the context for these numbers is not clear.

Thank you. We have tried to remove it accordingly. 

Introduction, line 61-62. I would suggest rewording as "In Ethiopia, previous studies assessing the prevalence of depression among PLWH have differed considerably, with estimates ranging from 11.7-76.7% [11-15]."

The comment is accepted and addressed in the introduction section line 61-63 accordingly. 

Introduction, line 71. Please change to "Depression has been associated with a wide range of factors, including age, sex ...”.

Thank you. We have tried to address it in the introduction section line 72-73 accordingly. 

Introduction, lines 78-81. Is this paragraph necessary? Although i completely agree that ending the HIV/AIDS epidemic is important, its direct link to depression requires more explanation, and is not directly relevant to the manuscript. So this section could be cut.

Yes it is necessary. Depression imposed substantial effects on the health of HIV positive patients. It leads to poor ART drug adherence and further transmission of HIV/AIDS. One of the seventeen goal of SDG is to end the HIV/AIDS epidemic by 2030. So, it is difficult to achieve SDG goal (end AIDS epidemic) in the presence of depression as it leads to poor ART drug adherence and further transmission of HIV/AIDS. We have tried to revise it in the introduction section line 81-83.

Methods, line 88. Please clarify that Dessie town is in Ethiopia, and perhaps indicate where Dessie is within Ethiopia (e.g. "A cross-sectional study was conducted between 1-30 March 2019 in public health facilities of Dessie Town, which is a city located within north-central Ethiopia").

Thank you. We have tried to clarify it in the method section line 91-94 accordingly. 

Methods, line 91. Should this be "The study population was randomly selected from all adults with HIV attending public health facilities...."

We have accepted the comment and addressed it in the method section line 96-97 accordingly. 

Methods, sample size. Do you mean that you assumed that 10.1% would have depressive symptoms (so that 89.9% did not)? Otherwise this prevalence seems a little high.

Thank you. It is editorial issue. It is to mean that 10.1% of patients without ART side effects have depressive symptoms.

Methods, lines 95-108. This section is perhaps a little long. It could be considerable shortened. For example: "The study sample size was calculated for the primary risk factor as presence of antiretroviral drug side effects, considered as a binary variable. Based on previous research [12], we wished to to detect differences in adjusted odds ratios of at least 4.7, assuming an underlying prevalence of 10.1%, power of 80% and 5% Type I error. After adjustment for 10% non-response, the minimum required sample size was 388. Systematic random sampling was used to select participants using the antiretroviral drug dispensary registration book as the sampling frame. Study participants were invited to participate when attending the health facility for antiretroviral drug collection."

Thank you. The comment is accepted and addressed in the method section line 100-107 accordingly. 

Methods, line 111. Please clarify if the card reviewed was the medical record?

Yes it was the medical record. We have tried to clarify it in the method section line 110 accordingly. 

Methods, line 114. Please consider rewording as "The questionnaire collected data on socio-demographic, clinical, psycho-social, and behavioral characteristics."

We have revised it in the method section line 113-114 accordingly. 

Methods, line 116. Please clarify that the PHQ-9 is used to measure depressive symptoms rather than depression directly. Please also clarify that the cut-off of 5 typically corresponds to mild, moderate or severe depressive symptoms as other cut-offs are sometimes used

Thank you. We used PHQ-9 scale to assess depressive symptoms. In the PHQ scale, a score of 0-4, represents no depression and a score of 5-9, 10-14, 15-19 and 20-27 represent mild, moderate, moderately sever and sever depressive symptoms respectively. In this study, PHQ score ≥ 5 was used as a bench mark to categorize depression. Accordingly, patients said to have depressive symptoms if they scored ≥ 5 and have no depressive symptoms if they scored <5.

Methods, line 125. Please give the median score (out of 10?) used to define "high" and "low" stigma. Also, where was this median obtained from, as it appears that only 30% had perceived stigma in Table 2 (instead of the 50% you would expect if the median was taken from this present study).

Stigma was assessed by using ten ‘yes’ or ‘no’ questions and the score ranged from 0 to 10. In our study, out of 380 study participants, 266 of them scored 0, 11 of them scored 1, 4 patients scored 2, 9 patients scored 3, 14 patients scored 4, 29 patients scored 5, 23 patients scored 6, 16 patients scored 7, and the remaining 8 patients scored ≥8 out of 10. The median score is the middle number in a given sequence of numbers when it’s ordered by rank. In this study, the median score is 0. Accordingly, having perceived stigma was defined if patients scored above the median score (i.e. above 0) which is 30%. Perceived stigma cannot be 50% since a score of greater than the median value was used to define perceived stigma. You can confirm it in manual calculation using the above frequency distribution. Median value can be calculated manually or using SPSS. 

Methods, line 131. as the study is cross-sectional, the independent variables are not really "predictors". Perhaps "explanatory variables", "factors" or "independent variables" would be better?

We have tried to use the term "independent variables" accordingly. 

Methods, line 135. As this is a cross-sectional study, it may not be appropriate to use the word "determinents". "associated factors" could be an alternative?

We have tried to use the term "associated factors" accordingly. 

Results, line 152. It would be good to give the overall prevalence of depression (for the whole population), before giving it in the sub-groups.

We have tried to put the overall prevalence of depression in the result section line 154 accordingly. 

Results, Table 3. Would it be possible to add the question used to define the variable "HIV sero-status disclosure" in a footnote to the table?

We have tried to put it as a table footnote accordingly. 

Results, Table 3. CD4 count, opportunistic infections and WHO stage are quite correlated - is it reasonable to include all in the model?

We have tried to re-analyze the data by removing ‘CD4 count’ and ‘WHO stage’ from the model accordingly. 

Discussion, line 227. Please say "times the odds" rather than "times the risk" here.

Although it is not clear, we have tried to revise it as “The odds of depressive symptoms were 2.6 times more among patients who took medication other than ART as compared to their counterparts”.

Discussion, line 237. You could consider adding the word all, so it reads "cannot be generalised to all people living with HIV".

The comment is accepted and addressed accordingly.

---

## [Editor Report · Decision Letter 2]

26 Jul 2021

Depressive symptoms and associated factors among HIV positive patients attending public health facilities of Dessie town: A cross-sectional study.

PONE-D-21-03982R2

Dear Dr. Damtie,

We’re pleased to inform you that your manuscript has been judged scientifically suitable for publication and will be formally accepted for publication once it meets all outstanding technical requirements.

Kind regards,

Colette Smith, PhD

Academic Editor

PLOS ONE
---

## [Editor Report · Acceptance letter]

28 Jul 2021

PONE-D-21-03982R2 

Depressive symptoms and associated factors among HIV positive patients attending public health facilities of Dessie town: A cross-sectional study. 

Dear Dr. Damtie:

I'm pleased to inform you that your manuscript has been deemed suitable for publication in PLOS ONE. Congratulations! Your manuscript is now with our production department. 

Kind regards, 

on behalf of

Dr. Colette Smith 

Academic Editor

PLOS ONE